# Constructing Pseudo-parallel Swedish Sentence Corpora for Automatic Text Simplification

**Daniel Holmer, Evelina Rennes**
Department of Computer and Information Science
Linköping University, Sweden
`firstname.lastname@liu.se`

## Abstract

Automatic text simplification (ATS) describes the automatic transformation of a text from a complex form to a less complex form. Many modern ATS techniques need large parallel corpora of standard and simplified text, but such data does not exist for many languages. One way to overcome this issue is to create pseudo-parallel corpora by dividing existing corpora into standard and simple parts. In this work, we explore the creation of Swedish pseudo-parallel monolingual corpora by the application of different feature representation methods, sentence alignment algorithms, and indexing approaches, on a large monolingual corpus. The different corpora are used to fine-tune a sentence simplification system based on BART, which is evaluated with standard evaluation metrics for automatic text simplification.

## 1 Introduction

Automatic Text Simplification (ATS) is a sub-field of natural language processing mainly focusing on the automatic transformation a text from a complex form to a less complex form, and in that way make texts accessible for weaker readers. Even though the modern ATS techniques vary in scale and efficiency, there is one constant; the need for large parallel corpora of standard and simplified text, in order to train the simplification system.

The acquirement of such corpora is however not an easy task. One theoretical option is to collect manually created simplifications, but that process is incredibly time consuming and often not feasible due to the enormous amount of text that is required by modern ATS systems.

A second option is to leverage already existing sources of parallel texts. One common example is the collection of articles from Wikipedia alongside their Simple Wikipedia counterpart. However, Xu et al. (2015) identified numerous problems to the dual Wikipedia approach, for example the fact the simple article most often is not a rewrite of the standard article. This can lead to a variation of the content in the articles that is large enough to make them unsuitable to be included in an aligned corpus. Moreover, the Simple English Wikipedia presents a limitation in text simplification research due to its sole availability in the English language. One way to overcome this problem is to translate the English texts into another language. For instance, Sakhovskiy et al. (2021) translated the WikiLarge dataset (Zhang and Lapata, 2017) into Russian.

Another possibility would be to follow the approach of Kajiwara and Komachi (2018), where a monolingual sentence corpus is divided into a standard and simplified part, and aligned with the best sentence matches between the two corpora. The result is a "pseudo-parallel" monolingual corpus; a parallel monolingual corpus that has been aligned with an unsupervised alignment algorithm rather than been manually constructed or collected from an already divided source, circumventing the previously mentioned problems. The approach was proven to perform well for both English and Japanese domains.

The aim of the work presented in this paper was two-fold. First, we aimed to create Swedish pseudo-parallel sentence simplification corpora[1] from a single monolingual Swedish sentence corpus. Second, we aimed to investigate how different methods and techniques used during the creation influence the performance of sentence simplification systems trained on the different cor-

---

[1]The corpora are made available at: `https://github.com/holmad/Constructing-Pseudo-parallel-Swedish-Sentence-Corpora-for-Automatic-Text-Simplification`

pora. The research question we explored was:

- For different alignment and embedding techniques, which alignment thresholds produce corpora that when used to fine-tune a BART model, produce sentence simplifications with the highest BLEU and SARI scores?

## 2 Related work

Data-driven approaches are common for most modern research in sentence simplification (Alva-Manchego et al., 2020). Data-driven does—in this context—refer to the collection of parallel corpora of standard-simple sentence pairs. These corpora are then used to train simplification systems by considering the simplification task as monolingual machine translation.

Much research has been conducted by exploiting the standard and simple versions of the English Wikipedia (Zhu et al., 2010; Coster and Kauchak, 2011; Woodsend and Lapata, 2011; Hwang et al., 2015; Zhang and Lapata, 2017). Additionally, the Newsela corpus (Xu et al., 2015) has been used for the creation of aligned corpora (Alva-Manchego et al., 2017; Zhao et al., 2018) , much alike Wikipedia. The Newsela corpus contains $1,130$ standard news articles, combined with up to five simplifications for each given article. The simplifications are created by professional writers, which overall should be an improvement in quality over the simplifications in the Simple English Wikipedia, which are produced by volunteers (Alva-Manchego et al., 2020). In a Swedish context, Rennes (2020) compiled a corpus of $15,433$ unique sentence pairs derived from the websites of Swedish authorities and municipalities. This comparatively small resource is the only available aligned corpus of standard-simple sentence pairs for Swedish.

In contrast to the previously mentioned corpora, which are based on alignment of sentences that are extracted from one source of standard sentences and another source of simplified sentences, the construction of a pseudo-parallel monolingual corpus includes the process of deciding if every given sentence should be considered as one of standard or less complexity. For this task, Kajiwara and Komachi (2018) calculated the, for English text widely used, Flesch Reading Ease Score (FRES) for each sentence, and in that way determined its complexity. The Swedish counterpart to FRES is called *Läsbarhetsindex* (LIX)

(Björnsson, 1968). Since LIX only measures the lengths of words, sentences, and ratios of long words, additional text complexity metrics have been developed for Swedish texts, such as the SCREAM (Falkenjack et al., 2013; Falkenjack, 2018) and SVIT (Heimann Mühlenbock, 2013) measures.

With MUSS, Martin et al. (2022) implemented a method to align paraphrases based on their similarity measures. In order to train a simplifier to produce simplifications, as opposed to just paraphrases, the authors employed ACCESS (Martin et al., 2020). ACCESS enables controllable output of sequence-to-sequence models by including special control tokens, that—among other things—can be used to limit the length of decoder output.

### 2.1 BART

BART (Lewis et al., 2020) is an autoencoder for pretraining models for sequence-to-sequence tasks. A BART model is trained by inputting text corrupted with a noising function, and learning to reconstruct the text to its original state. Hence, it is a *denoising* autoencoder. BART utilises a bidirectional encoder[2], where random tokens are masked and the document is encoded by considering tokens in both directions. For the prediction of the masked tokens, each token is predicted independently by considering the entire input sequence. Since text-generation is a task that only considers the current and previous input, a standard BERT model is unsuitable for text generation[3] (Lewis et al., 2020). With BART, the bidirectional encoder is paired with an auto-regressive left-to-right decoder. The auto-regressive decoder predicts tokens by considering the current token combined with the leftward context, and can therefore generate new text.

The combination of the two components allows BART to apply any noising function, compared to previous autoencoders that are tailored for a specific function (Lewis et al., 2020). The number of possible pre-training tasks that can be employed by BART is therefore also significantly larger than, for example, BERT.

---

[2]The structure is very similar to that of BERT (Devlin et al., 2019), but some discrepancies can be noted. For instance, BART replaces ReLU with GeLU activation functions. See Lewis et al. (2020) for details.

[3]However, the weights of a BERT model can be used in a warm-start procedure of an encoder-decoder model to achieve similar capabilities. See for example Rothe et al. (2020) and Monsen and Jönsson (2021)

## 3 Data

We used several different datasets for different tasks. Table 1 provides an overview of the datasets used.

**The Stockholm-Umeå Corpus** (or *SUC*) (Gustafson-Capková and Hartmann, 2006) is a balanced corpus of Swedish texts from the 1990s. The style of text is varied, and it is sometimes used as a baseline for standard use of the Swedish language during the time period (see for example Pettersson and Nivre (2011)). In total, the corpus consists of $1,166,593$ tokens and $74,245$ sentences.

**The NyponVilja dataset** consists of OCR scans of books from Sweden's largest publisher of easy-to-read books, *Nypon och Vilja Förlag*, targeting children and youths. Each book is graded by human experts with a readability level, where level 1 denotes a book that is the easiest to read and level 6 denotes books that provide the most challenge for the readers.

**The CCNET dataset** is provided by Common Crawl[4], a non-profit organisation that uses web crawlers to collect an enormous amount of text data from all around the web, and makes it freely available to the public. The organisation collects and publishes a new data snapshot approximately 10 times a year[5], each snapshot in the size range of $\approx$ 100–300TB whereof 20–30 TB is raw text data.

We used the Swedish part of the CC-100 dataset, previously used to recreate the training of XLM-R (Conneau et al., 2020), for the sentence alignment task. The dataset was created by researchers at StatMT[6], by applying the CC-Net pipeline to extract datasets for 100 different languages from the Common Crawl snapshots created during the time period January–December 2018. The Swedish dataset comprises 80GB uncompressed text, in the form of $580,387,314$ paragraphs. From these paragraphs, $61,959,899$ sentences were extracted for further pre-processing and annotation.

The data was further prepared for alignment by roughly following the procedure in Raffel et al. (2020). However, an additional step was introduced to rearrange the data from paragraphs to sentences. This step was added since the task is to align sentences, not paragraphs. It was there-fore also necessary to annotate the dataset on the sentence level.

We used the SAPIS (Fahlborg and Rennes, 2016) pipeline to tokenise each sentence with Efselab (Östling, 2018), and to annotate each sentence with a subset of the SCREAM metrics previously identified by Santini et al. (2020).

**PK18** (Lindberg and Kindberg, 2018) is a corpus totalling 1,005 texts pairs. Each pair consist of an original version of the given text, and a simplified version of the same text. The texts origin from four Swedish organisations and municipal-, regional- and state departments; *Riksförbundet för utvecklingsstörda barn, ungdomar och vuxna (FUB)*, *Linköpings Kommun*, *Region Östergötland*, and *Specialpedagogiska myndigheten (SPSM)*. The simplified versions were written by experts, and were manually aligned with the corresponding original version of the texts.

PK18 is currently the largest available corpus suitable for use as a gold standard for the evaluation of Swedish ATS systems. Since this work focused on sentence-level simplification, only the pairs aligned in a 1-1 manner were used. The result was a dataset of 467 sentence pairs, with the purpose of being used as the test dataset for the fine-tuned text simplification system.

## 4 Implementation

This section describes the creation of the pseudo-parallel corpora and their usage in text simplification systems. The procedure can be outlined in four steps. First, the sentences were classified as being of either standard or simple complexity. Second, the sentences were aligned. Third, the different corpora were provided as training data to fine-tune multiple text simplification systems. Finally, the performance of each of the systems was assessed with standard evaluation metrics.

### 4.1 Labelling of sentences as standard or easy

Following Kajiwara and Komachi (2018), the sentence dataset was divided into two subsets, one with standard sentences and one with easy sentences.

We used a classification model to determine if the sentences from the CCNet dataset should be seen as "standard" or "easy". The model was realised with the implementation of Support Vec-

---

[4]https://commoncrawl.org

[5]Each snapshot can be found at https://index.commoncrawl.org

[6]https://data.statmt.org/cc-100/

| Dataset name | Sentences | Tokens | Usage |
|---|---|---|---|
| SUC | 74,243 | 1,166,593 | Standard sentences used for training of the SVM sentence classifier. |
| NyponVilja | 54,938 | 459,540 | Easy sentences used for the training of the SVM sentence classifier. |
| CCNet subset | 61,959,899 | 832,996 921 | Sentences which were classified as either easy or standard, and then aligned to form the easy/standard sentence pairs of the pseudo-parallel corpora. |
| PK18 subset | 467 (sentence pairs) | 7,873 (standard) 6,429 (simplified) | A manually annotated dataset that is used for evaluation of the sentence simplifier trained on the aligned corpora. |

Table 1: Overview of the different datasets used.

tor Machine (SVM) found in the Python library scikit-learn (Pedregosa et al., 2011). We annotated each sentence with a subset (described in Santini et al. (2020)) of the text complexity metrics from SCREAM (Falkenjack et al., 2013; Falkenjack, 2018), previously known to predict text complexity in Swedish. Since the metrics vary in scale (for instance, some metrics are ratios while other are raw frequencies), they were standardised by removing the mean and scale to unit variance, before being used as features to represent a sentence in the SVM.

The SVM was then trained with the standard sentences (from SUC) and the easy sentences (from NyponVilja) as class labels. A 10-fold cross-validation process was applied to evaluate the model performance. Averaged over all folds, the SVM classifier performed with an F1-score of 82%. This SVM classifer was then used to assign all sentences from the CCNet dataset as of either standard or easy complexity.

## 4.2 Alignment of sentences

The alignment of sentences labelled in the previous section can be divided into two categories: alignments based on similarities of individual word embeddings between sentences, and alignment based on the similarity of embeddings of whole sentences.

A common functionality between the two approaches is the ability to filter the resulting corpus with regard to the alignment threshold. A higher threshold would allow fewer sentence pairs to be included in the corpus, but the pairs that were included would be more similar according to the cosine distance, and therefore probably of higher quality. Inversely, a lower threshold would include more sentence pairs, but their similarity would on average be lower. To investigate this trade-off, corpora with the alignment threshold of both 0.8 and 0.9 were created.

### 4.2.1 Word-based embeddings

At its core, this approach is based on the method originally proposed by Song and Roth (2015) and later used by both Kajiwara and Komachi (2018) and Rennes (2020), where sentences were aligned according to their similarity at the word level. Different alignment algorithms were used to perform the task, where Kajiwara and Komachi (2018) implemented *Average (AA)*, *Maximum (MA)*, and *Hungarian (HA)* alignment algorithms, as well

as the *Word Mover's Distance (WMD)*. Rennes (2020) implemented the *AA*, *MA*, and *HA* alignment algorithms.

The main difference in this work when compared to the aforementioned works is the much increased dataset size; an increase of several million sentences. This brings forth some additional challenges, mainly regarding the computational complexity during the alignment process. For this reason, we only used the *AA* and *MA* algorithms. Both *HA* and *WMD* resulted in a dramatic increase in the required computations, which were not feasible to perform given the available hardware and time frame.

Average alignment similarity (AAS) calculates the pairwise cosine similarities between all the words of sentence *x* and sentence *y* and averages them over the number of pairs (see Equation 1).

$$AAS(x, y) = \frac{1}{|x||y|} \sum_{i=1}^{|x|} \sum_{j=1}^{|y|} cos(x_i, y_j) \quad (1)$$

Maximum alignment similarity (MAS) can be seen as a refinement of the AAS, since it does only take into account the best (maximum cosine similarity) word pair between sentence *x* and sentence *y* (see Equation 2).

$$MAS_{asym}(x, y) = \frac{1}{|x|} \sum_{i=1}^{|x|} \max_{j} cos(x_i, y_j) \quad (2)$$

Equation 2 describes an *asymmetric* similarity, meaning that there will be different total similarity scores depending on if each of the words of sentence *x* gets paired with its maximum similarity in sentence *y*, and vice versa[7]. Therefore, to get a symmetric MAS, we add the averages of the asymmetric MAS(*x*, *y*) and MAS(*y*, *x*), as described in Equation 3.

$$MAS(x, y) = \frac{1}{2}MAS_{asym}(x, y) + \frac{1}{2}MAS_{asym}(y, x) \quad (3)$$

In earlier works, MAS has shown to be well performing, and the alignment algorithm of choice of both Kajiwara and Komachi (2018) and Rennes (2020).

---

[7]Unless the sentences are identical, but that would of course make the whole alignment procedure unnecessary

Another consequence of the increased amount of data is the need to restrict the search problem during the alignment process. Even though only the computationally least demanding alignment algorithms were used, to calculate the cosine distances in a *N:M* manner (that is, between every easy sentence and every standard sentence) would be too computationally expensive. Therefore, a more efficient method of calculating the similarities was implemented.

We used MinHashLSH[8] to construct an index from the easy sentences, and query the index with the standard sentences to create a mapping of potential sentence pairs for alignment (see step 1, 2, and 3 in Figure 1). MinHash allows the matching of sentences that share fewer features in the syntactic sense, than for example SimHash as proposed in earlier works, but still set a requirement that the sentences have to share similarities at a given threshold. For this work we used the Jaccard similarity of `0.5` for a sentence pair to be considered a possible match. This allowed for a relatively large range of possible matches, but still dramatically reduced the search space. The index was constructed with a feature window of `5` and the `num_perm` parameter of `16`.

After the construction of the index and the extraction of possible matching sentences, we used Fasttext[9] pre-trained Swedish word vectors to embed every word in every sentence of the matching pairs. In order to reduce the memory footprint of the vectors, we reduced the dimensions from the default `300` to `100` dimensions. This allowed for more vectors to be loaded in memory, and allowed larger batches of computations of several sentences at once. This significantly improved the computational overhead for the alignment module. The embeddings of the words in the matched sentences then got passed to the alignment module (see steps 4, 5, and 6 in Figure 1).

### 4.2.2 Sentence-based embeddings

For this approach, each sentence was represented as a sentence embedding via Swedish sentence-BERT (Rekathati, 2021). Each embedding from the standard bucket was compared to all of the embeddings from the easy bucket, and the pair of standard and easy sentences with the highest

---

[8]from the datasketch package http://ekzhu.com/datasketch/lsh.html
[9]https://fasttext.cc/docs/en/crawl-vectors.html

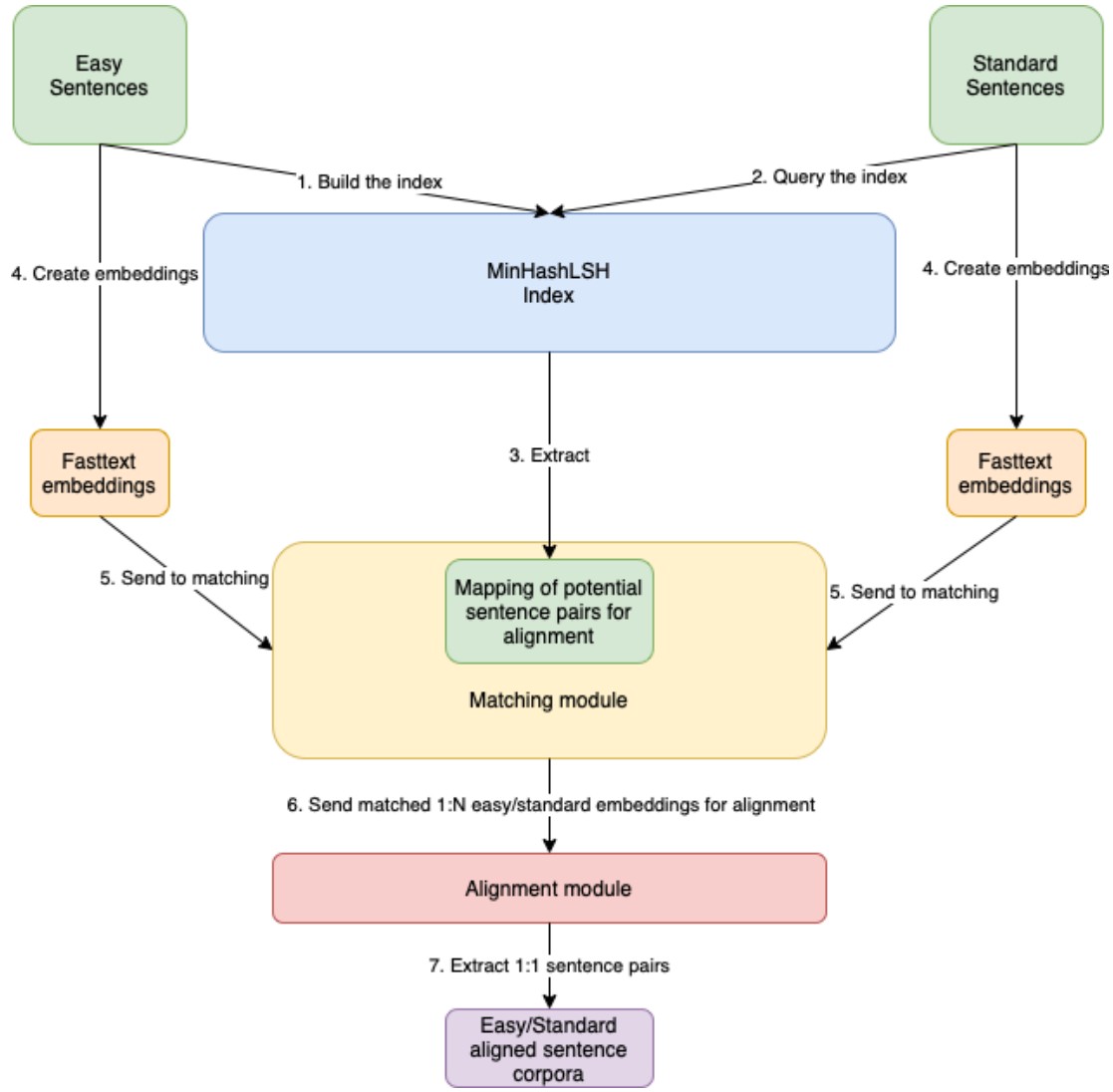

Figure 1: High-level overview of the alignment of sentences with the word-level Fasttext embeddings.

cosine similarity was aligned and added to the corpus. To speed up this process and forgo the quadratic complexity of an exhaustive search, the embeddings of the easy sentences were indexed using Faiss (Johnson et al., 2019). Since Faiss requires all embeddings to be loaded into memory when constructing the index, we employed PCA to reduce the output dimension of the sentence transformer model from 768 to 128[10]. The slight reduction in quality for each embedding was deemed to be outweighed by the ability to use all easy sentence embeddings for the index training and construction.

For this work, we used the `IVFFPQ`-index from

Faiss, which utilises both coarse- and fine quantisation to reduce both search times and index disk size. The index was trained with the parameters `nlist=2048` (the number of Voronoi cells), `nbits=8` (the number of bits to represent the codes per each subvector), and `M=8` (the number of subvectors per vector). Additionally, each embedding vector was normalised to support measuring cosine distances as opposed to Euclidean distances.

For the standard sentences, each sentence embedding was queried to the index, and the easy sentence with the highest cosine similarity to the queried standard sentence was extracted if it adhered to the given similarity threshold set for the current corpus.

[10]This process was based on the following code from SentenceTransformers https://github.com/UKPLab/sentence-transformers/blob/master/examples/training/distillation/dimensionality_reduction.py

| Embed-ding type | Word align-ment algo-rithm | Threshold | Sentence pairs | Avg. sentence length (easy) | Avg. sentence length (stan-dard) | BLEU | SARI |
|---|---|---|---|---|---|---|---|
| - | - | baseline | - | - | - | 22.81 | 12.80 |
| word | AA | 0.8 | 440,259 | 8.24 | 12.76 | 10.17 | 33.11 |
| word | AA | 0.9 | 40,014 | 7.16 | 8.05 | 17.29 | 28.31 |
| word | MA | 0.8 | 442,152 | 8.25 | 12.77 | 9.53 | **33.24** |
| word | MA | 0.9 | 40,017 | 7.16 | 8.05 | 16.71 | 29.51 |
| sentence | - | 0.8 | 6,560,372 | 7.09 | 12.25 | 4.04 | 30.43 |
| sentence | - | 0.9 | 652,964 | 6.23 | 9.20 | 3.64 | 30.29 |

Table 2: The created corpora and their evaluation scores when used to train the simplification system.

## 4.3 Simplification module

Each corpus was used to fine-tune a simplifier based on a Swedish BART model[11], developed by KBLab. They pre-trained the model on approximately 80GB of text (around 15B tokens) with the help of Fairseq[12], and subsequently converted it to be compatible with the Huggingface Transformers Python-library (Wolf et al., 2020). The pre-trained model consisted of approximately 139M parameters.

In our work, the fine-tuning and evaluation pipeline was in large part built with the Transformers library. Each sentence pair were tokenised using the pre-trained model's tokeniser with the `AutoTokenizer` class and the model was loaded using the `AutoModelForSeq2SeqLM` class. For the fine-tuning, the hyperparameters were consistent for all models, with the `learning rate=3e-05` and `batch size=32`. Furthermore, the number of `warmup steps` were `500` and the `weight decay=0.1`. The optimisation algorithm was the default `AdamW` and each simplification model was fine-tuned for between `1` and `10` epochs, depending on corpus size. In general, the hyperparameters were kept close to the default values, and the ones we experimented with only showed minor differences in performance.

From each corpus, 90% of the sentence pairs were used as training data, and 10% were used as validation data.

---

[11]https://huggingface.co/KBLab/bart-base-swedish-cased
[12]https://github.com/facebookresearch/fairseq

## 4.4 Evaluation

For the evaluation, we applied two metrics commonly used for the assessment of ATS systems – BLEU and SARI. BLEU (BiLingual Evaluation Understudy) (Papineni et al., 2002) is calculated with modified unigram precision and a brevity penalty factor between a target and reference sentence. The SARI metric (Xu et al., 2016) compare **s**ystem output **a**gainst **r**eferences and against the **i**nput sentence. The purpose of SARI is to quantify the simplification of sentences based on words that are *added*, *deleted*, or *kept* by the simplification system. (Alva-Manchego et al., 2020) describes the intuition behind SARI as that the system is rewarded for the *addition* of n-grams that occur in any of the references but not in the input, the *keeping* of n-grams both in the output and the references, and the avoidance of over-*deleting* n-grams.

Unfortunately the PK18 subset is limited by its small size, but it is to the best of the authors knowledge the only manually aligned simplification dataset in Swedish, and future studies would benefit from a larger, high quality dataset. For this study, we did however use the PK18 subset of 467 manually aligned sentence pairs to evaluate the performance of the BART simplifiers trained on the different generated corpora. Each sentence pair was passed as test data, and BLEU and SARI metrics were calculated. As a baseline, we calculated the BLEU and SARI metrics for the test dataset when no simplification was performed (i.e. the original sentence was used as the system output sentence and the gold standard simplified sentence was used as the reference sentence). For both BLEU and SARI calculations,

we used the implementation from EASSE (Alva-Manchego et al., 2019).

## 5   Results

For simplicity, the created corpora are referred to with the notation of *[embedding type]_[word alignment algorithm]_[alignment threshold]*. For example, the corpus in the second row of Table 2 is referred to as `word_AA_0.8`.

In Table 2, the results of the corpora created with the alignment and embedding methods described in Sections 4.2.1 and 4.2.2 are presented. The baselines for BLEU and SARI were calculated as described in Section 4.4 (i.e. they were calculated as if no simplification was conducted at all).

All of the corpora performed better than the baseline SARI. However, the best performance was shown by both word embedding-based corpora with a filtering threshold of 0.8, with a SARI score of over 33. This is higher than the corpus aligned with the help of sentence embeddings, which had a SARI of 30.43. Of the two best-performing word embedding-based corpora, the one aligned with the MA algorithm performed with a slightly higher SARI score than the AA one.

For the BLEU score all of the corpora showed lower values than the baseline. The corpus based on word embeddings and with an alignment threshold at 0.9 did however show BLEU scores fairly close to the baseline. The rest of the corpora performed significantly lower.

It is clear that the number of sentence pairs is closely related to the alignment threshold. For all embedding type/word alignment algorithm combinations, the corpus with a higher threshold also consisted of fewer sentence pairs than their lower threshold counterparts.

## 6   Discussion

In this section the results for the different conducted experiments will be discussed.

### 6.1   Alignment results

Inspecting the results in Table 2, a first thing to note is that all of the models fine-tuned on the corpora performed with higher SARI scores than the baseline. Furthermore, the two corpora created using embeddings on the word level and the sentence alignment threshold of 0.8, `word_AA_0.8` and

`word_MA_0.8`, showed the highest SARI scores (33.11 & 33.24) in this study.

On the other hand, the `word_AA_0.9` and `word_MA_0.9` corpora showed significantly higher BLEU scores than the rest, while at the same time exhibiting relatively low SARI scores. One explanation for this behaviour is that the simplifications from the models fine-tuned on these corpora often include only minor changes to the original sentence. In some cases, no change from the original sentence can be observed at all. As a consequence, since few (or none) *add*, *delete*, or *keep* operations can be rewarded, the SARI score will be kept low. Inversely, the similarity between the original and output sentences will benefit the BLEU score. The evaluation dataset contains, in many cases, small differences between the standard and simplified sentence, with only small parts of information either added or deleted. This in turn leads to a situation where the reference and original sentences are so similar that a (relative to the baseline) high BLEU can be achieved by just keeping the original sentence.

When looking at both the corpora based on sentence embeddings (`sentence_0.8` and `sentence_0.9`), it can be noted that the SARI scores are somewhat average compared to the other corpora. The BLEU scores are however significantly lower. One possible explanation for this behaviour could be that BLEU is more restrictive than SARI, in the sense that the same n-gram have to be present in both the target and reference sentence for BLEU. Since the sentence embeddings are a semantic representation of the sentence, two sentences could have high similarity scores on the sentence level while having a low ratio of shared n-grams.

Overall, the `word_MA_0.8` corpus performed with the most balance between the BLEU and SARI scores, closely followed by `word_AA_0.8`.

### 6.2   Evaluation metrics

While BLEU has been used as a metric for the evaluation of automatic text simplification systems, it is problematic to use. Sulem et al. (2018) showed how BLEU fails to serve as a useful evaluation metric for sentence splitting operations. Since the corpora created in this work are aligned in a sentence-to-sentence manner, this point is of less importance for this specific evaluation. However, the authors did also find that BLEU of-

ten *negatively* correlates with simplicity, and may penalise simpler sentences instead of rewarding them. To rely on BLEU as the only metric for evaluation is therefore not to recommend. In this work, its main purpose is instead to indicate the *similarity* of the reference and system output, not necessarily the difference in *simplicity*. For example, the BLEU metric gives support to the observations that the simplified sentences of the models fine-tuned on word_AA_0.9 and word_MA_0.9 in many instances is just a cut-off version of the standard sentence, where either the beginning or end of the sentence have been removed. For this particular behaviour, the BLEU metric provided valuable information despite its other apparent flaws in the task of text simplification.

Another thing to note is the low BLEU scores overall, but in particular for the corpora based on sentence embeddings. The low overall scores can probably, as also observed by Kajiwara and Komachi (2018), partly be attributed to the lack of multiple reference sentences in the test dataset. An additional contributing factor to low scores for the corpora based on sentence embeddings is probably the behaviour that sentences with named entities often get aligned with sentences containing completely different entities. This leads to a corpus of sentences with a lower ratio of exact word-to-word matches. When evaluating simplification models fine-tuned on these corpora, the BLEU metric would probably be more affected by this than the SARI metric.

In recent years, much of the published research on text simplification systems has used SARI as an evaluation metric. One of its main merits is that it is good at assessing a system's ability to perform lexical paraphrasing. (Alva-Manchego et al., 2021) suggest using a combination of multiple metrics to capture different aspects of text simplification. In future studies it would be interesting to implement a wider array of metrics, for example BERTScore (Zhang et al., 2020) or METEOR (Denkowski and Lavie, 2011), to further examine the quality of the corpora.

## 7 Conclusion

The aim of the work presented in this paper was to create a Swedish pseudo-parallel sentence simplification corpus from a single monolingual Swedish sentence corpus, and to investigate how different methods and techniques used during the creation influence the performance of sentence simplification systems trained on the different corpora.

From the results, it can be seen that the model fine-tuned on a corpus created with word-based embeddings, the Maximum Alignment algorithm, and an alignment threshold of 0.8 performed with the best SARI and acceptable BLEU scores. It is however unclear how much the different indexing methods impacted the performance of the alignment process, and exactly how the quality of the corpora was affected.

Both the investigated methods of creating pseudo-parallel corpora for sentence simplification show promising results. Future studies should conduct a further investigation on different parameter choices, mainly when constructing the indices to help the alignment process, and explore how they impact the quality of the corpora. The resulting corpora should also be thoroughly evaluated with regard to different aspects of text simplification; with a combination of qualitative evaluations, additional evaluation metrics, and a larger test dataset.

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
