# OpenReview forum: "Constructing Pseudo-parallel Swedish Sentence Corpora for Automatic Text Simplification"
_NoDaLiDa/2023/Conference — NoDaLiDa 2023_

### Official Review · Reviewer_DQvf · 2023-03-01
**Constructing Pseudo-parallel Swedish Sentence Corpora for Automatic Text Simplification**

**Rating:** 8
**Confidence:** 5

**Review:**

The paper presents experiments on automatic text simplification for Swedish using synthetic training material extracted from monolingual corpora. The work is extensive and well presented. The idea is to align sentences based on word and sentence embeddings from examples marked as standard or simplified. The model itself is based on Swedish BART, a pre-trained encoder-decoder model, which is then fine-tuned for the task on the pseudo-parallel data.

The challenge is the complexity of alignment and the authors compare word-level with sentence-level approaches and evaluate them om a small test set of annotated sentence pairs. the results show that the word-level model seems to outperform the sentence-level method, which surprises me. The test set is very limited and, unfortunately, it is not easy to draw explicit conclusions from the results but the experiments are certainly comprehensive and well presented. The study is well conducted and worth to be published.

Some more detailed comments:
- the introduction presents some motivations why creating pseudo-parallel data sets is interesting. I am not sure whether I understand the difference between option one and two. What does it mean to manually collect parallel data and how is it different from option two (collect existing parallel texts)? Do you mean manually translating texts into simplified forms in option one? You also mention problems with simplified Wikipedia but isn't the main problem that simple Wikipedia doesn't exist in most languages? Does it exist for Swedish?
- maybe you should also mention yet another option: translating simpified data (e.g. Wikipedia) to align to standard language alternatives
- you mention that the unsupervised alignment method is language-agnostic based on a study that looks at only two languages (Japanese and English). That's bit quick in jumping to say that this proves to be language agnostic. The claim should be reduced to something less dramatic
- in part two you mention that assessing every sentence with some complexity metric in a large corpus is cumbersome - but isn't that exactly what you also need to do in your approach in order to divide sentences into the two buckets (standard or simplified). I don't understand the argument.
- the next sentence there also explains the approach that removes the filtering step, which is strange as there is no procedure with steps that you mention before. From what kind of procedure is this removing a step? Furthermore, it does not look like the filtering step is removed after all, because even in that approach the control tokens need to be created based on some metrics that indicate simplification and this is in some sense equivalent of being the filter in the end (but I kind of understand your point)
- the introduction of BART is fine but I still find it interesting that nowadays encoder-decoder models are presented as a combination of LMs like BERT and GPT, whereas this architecture is basically the original transformer architecture proposed for neural MT long before BERT and GPT. I would rather say that BERT and GPT are based on the original encoders and decoders of the original NMT transformer model. So, simply saying that BART uses the original encoder-decoder transformer model would do it for me but the hype around LLMs seems to make people forget about the history.
- with respect to evaluation, I was wondering how reliable the results are for such a small data set. 467 sentence pairs are really not much and I would like to see some discussion on the confidence we can have in the scores you present. I would not trust the numbers very much, especially also with BLEU scores below 10 - what does that actually tell us? A matching punctuation and some sequences of tokens? But BLEU is anyway not a good metric here as you also discuss in the paper.
About the hash-index in the word-level alignment: I understand the need for reducing the search space but I still wonder how much the requirement for surface-form similarities is limiting the approach. I would expect that a lot of interesting simplification examples are ruled out in this way. Could it be an artifact of the test data that the word-level approach is working well because a lot of surface-form similarities can be seen in those examples? Could you add some more qualitative analyses of individual examples in the test set to see whether there is some trend like this?
- final point: the baseline is really a stupid baseline - that's more like a sanity check comparison. Is there some other kind of baseline that one could use? However, I don't have any suggestion.
- Good that you added a discussion on the problems with BLEU, but i wonder why you even use BLEU considering all the issues. Would there be other metrics that are more suited like METEOR or other metrics that at least allow synonym substitution?

**Paper Type:**

Long paper

---

### Official Review · Reviewer_3fCx · 2023-03-10
**A new corpus for automatic text simplification**

**Rating:** 9
**Confidence:** 4

**Review:**

The paper presents a new, large monolingual corpus containing Swedish pseudo-parallel sentence pairs for the task of automatic text simplification.

Strengths: A large dataset/corpus complied from different sources is necessary for training and evaluating state-of-the-art models. The paper is well-structured. It contains a detailed description of the workflow, there is an exact description of the classes and hyper-parameters that were used in the evaluation, which was also was performed systematically.

Weaknesses: It is not clear what is the quality of the datasets that the corpus was complied from.

Detailed comments:

Data section: Can you say something about the quality of the different datasets?

l. 254: 'Readability level', who is the target reader?

l. 354: What is the difference between 'standard' and 'simple' complexity?

l. 377: The metrics were not introduced, therefore it is not clear what does 'removing the mean and scale to unit variance' means.

Figure 1: Consider enlarging the font size inside the figure.

l. 589: Please explain 'normalised and queried to the index.'


**Paper Type:**

Long paper

---

### Official Review · Reviewer_Bya4 · 2023-03-17
**Swedish sentence simplification dataset whose quality is difficult to assess**

**Rating:** 5
**Confidence:** 3

**Review:**

The paper describes the work of building pseudo-parallel Swedish sentence corpora, aligned on sentence level, and the training of several models/systems on this dataset. First sentences are classified as standard or simple, then the sentences are aligned in terms of similarity using either word embedding matching (two variants) or sentence embedding matching, as well as different embedding similarity thresholds, resulting in the different corpora. A BART-based model is fine-tuned on the different corpora with the aim of finding the representation and aligning approach that is best suited.

 * Line 014 - Is there a word missing here?

 * Some of the abbreviations used are not properly introduced with their full-form before used, such as BART, BERT, BLEU and SARI.

 * Descriptions of the BLEU and SARI metrics are missing.

 * Section 4.1 - Please also mention how the text is represented when given to the SVM classifier (TF, TF-IDF, or Boolean vectors, etc).

 * Section 4.2 - It is not clear to me how you reduce the dimensionality of the BERT embeddings form 768 to 128. Please explain.

 * Line 785 - I am curious to hear why you think the BLEU scores are so low for the last two corpora.

 * Section 6.2 - I think a manual evaluation would have really strengthened this study. You argue that the BLEU scores are not reliable, which also the baseline indicate (and I am unclear about your opinion when it comes to SARI). Thus I think a manual analysis of a sample would have been great for understanding what is going on here. This could, e.g., be one sample from the top word embedding approach/system and one from a sentence embedding approach/system. Right now, as someone not too familiar with sentence simplification, I am somewhat questioning the quality, and thus usefulness, of this dataset.

 * Section 6.2 - Currently the discussion seems a bit one-sided, SARI is basically not mentioned.

 * Please specify if, and possibly where, the dataset will be made available.


Pros:
 * Automatic text simplification dataset for Swedish.
 * Different sentence alignment approaches are explored.

Cons:
 * Difficult to interpret the results based on the performance metrics used and their descriptions (and following discussion).
 * A manual evaluation would be very useful.
 * See above comments.


**Paper Type:**

Long paper

---

### Decision · Program_Chairs · 2023-03-17

Accept